# FURINA: FREE FROM UNMERGEABLE ROUTER VIA LINEAR AGGREGATION OF MIXED EXPERTS

## ABSTRACT

The Mixture of Experts (MoE) paradigm has been successfully integrated into Low-Rank Adaptation (LoRA) for parameter-efficient fine-tuning (PEFT), delivering performance gains with minimal parameter overhead. However, existing MoE-LoRA methods suffer from a critical limitation: their reliance on discrete routers prevents integration of MoE components into the backbone model, resulting in persistent computational overhead and increased system complexity during inference. To address this challenge, we propose **FURINA**, a novel **F**ree from **U**nmergeable **R**outer framework based on **LIN**ear **A**ggregation of experts. To the best of our knowledge, FURINA represents the first fully mergeable MoE-LoRA method that can be seamlessly reparameterized into the backbone model after training. This enables FURINA to function as a plug-and-play component within any LLM deployment framework–a capability where standard MoE-LoRA approaches fail–while maintaining equivalent learning capacity. FURINA introduces a Mergeable Self-Routing mechanism that leverages angular similarity between inputs and adapter directional components to activate experts, which are then scaled by the shared magnitude vector. This design enables the output norm to naturally reflect expert importance, facilitating both router-free operation and seamless merging. The expert selection loss further enhances this behavior by encouraging sparsity and alignment with standard MoE activation patterns. Additionally, we incorporate a shared expert within the MoE-LoRA block to provide stable, foundational knowledge. Extensive experiments across 9 benchmarks and 3 different LLMs demonstrate that FURINA significantly outperforms standard LoRA while matching or surpassing existing MoE-LoRA methods, all while eliminating their additional inference-time overhead. We plan to open-source our implementation upon publication.

## 1 INTRODUCTION

While Large Language Models (LLMs) have achieved remarkable success, effectively balancing their formidable performance with stringent resource constraints remains a significant challenge. Parameter-efficient fine-tuning (PEFT) methods, particularly Low-Rank Adaptation (LoRA), have emerged as a dominant solution, enabling effective adaptation of LLMs. Inspired by the success of Mixture-of-Experts (MoE) architectures like Mixtral 8×7B (Jiang et al. (2024)), recent work has sought to integrate MoE principles into LoRA-based PEFT to enhance model capacity within a fixed parameter budget. For instance, MixLoRA (Li et al. (2024)) directly introduces the Mixtral-styled MoE into the LoRA approach. LoRAMoE (Dou et al. (2024)) proposes to alleviate knowledge on specific LoRA adapters, and SLIM (Han et al. (2025)) introduces identity layers and a dynamic merging strategy to leverage the pre-trained knowledge of the backbone model effectively. Although integrating MoE with LoRA improves expressivity, it inherits a critical limitation: the introduction of an individual routing function prevents the adapters from being merged back into the base model. This fundamentally undermines a core advantage of standard LoRA, as the unmerged experts necessitate specialized, often inefficient, inference logic. Consequently, these methods suffer from limited support in high-performance inference frameworks like vLLM (vLLM Team) and incur unavoidable computational overhead, posing a major barrier to their practical deployment.

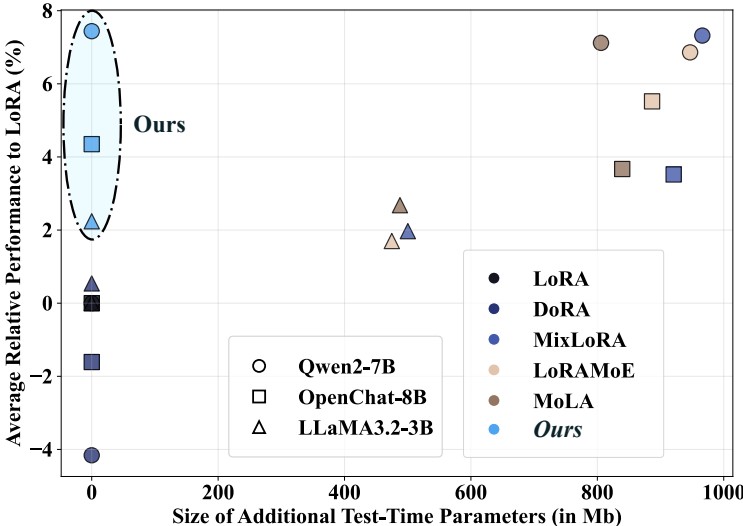

Figure 1: Relationship between additional test-time parameters and relative performance gain compared to standard LoRA. FURINA achieves comparable or superior performance gains to standard MoE-LoRA approaches while sharing the same architecture with standard LoRA during inference, resulting in up to 1.5× speedup.

To overcome these limitations while preserving the performance benefits of MoE, we propose FURINA (**F**ree from **U**nmergeable **R**outer via L**IN**ear **A**ggregation), a novel fully mergeable MoE-enhanced LoRA architecture which matches the performance of the standard MoE–LoRA methods while overcoming the deployment complexity and overhead. The core of our approach is a Mergeable Self-Routing mechanism, which replaces the traditional router with three key components: (1) Decoupled Learning of Direction and Magnitude: decouples the LoRA adapters by applying column-wise normalization to the weight matrices, isolating their directional components. (2) Shared Learnable Magnitude Vector: introduces a single, shared magnitude vector that scales the outputs of all experts uniformly, ensuring the norm of an expert's output directly reflects its activation strength. (3) Expert Selection Loss: employs a loss function that encourages sparse, divergent expert activation by maximizing the contribution of the most relevant experts. Specifically, the input is first projected by the normalized LoRA matrices to produce normalized logits, then scaled by the shared magnitude vector. This design allows the norm of each expert's output to naturally represent its relevance to the input, enabling dynamic, router-free routing. We also introduce a Shared Expert (SE) within the MoE–LoRA block. This expert provides foundational knowledge across the data corpus without introducing non-linearities that would prevent merging. By integrating these modules, FURINA seamlessly transitions between two phases: during training, it operates as a full, capacity-enhanced MoE architecture; during inference, the experts and the shared expert are linearly aggregated and can be merged into a single LoRA adapter or directly into the backbone model, introducing zero overhead. A comparative summary of FURINA against full fine-tuning and other PEFT methods is provided in Tab. 1.

Table 1: Comparison of the proposed FURINA with different fine-tuning strategies

| Method | PEFT | MoE During Training | Zero Extra Test Time Cost |
|---|---|---|---|
| Full-SFT | × | × | ✓ |
| LoRA (Hu et al. (2022)) DoRA (Liu et al. (2024b)) | ✓ | × | ✓ |
| MixLoRA (Li et al. (2024)) LoRAMoE (Dou et al. (2024)) MoLA (Gao et al. (2025)) SLIM (Han et al. (2025)) | ✓ | ✓ | × |
| FURINA (Ours) | ✓ | ✓ | ✓ |

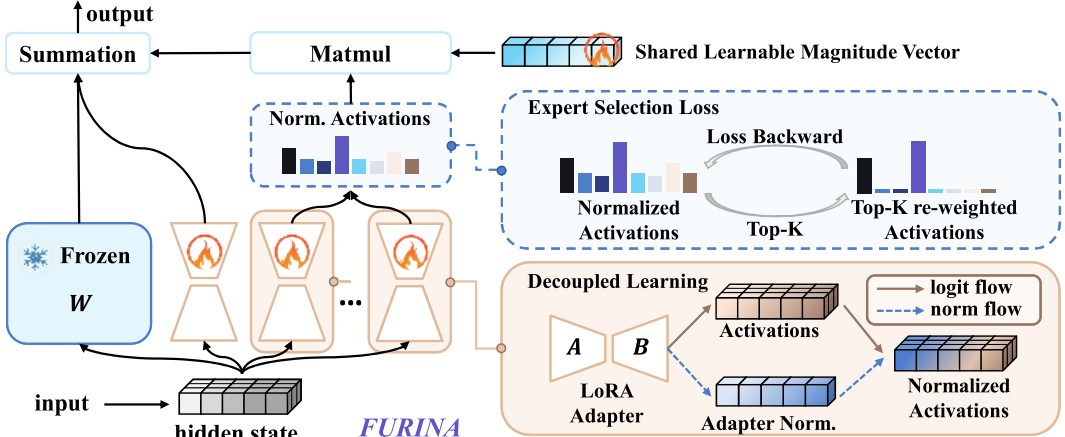

Figure 2: The overall framework of FURINA during training. LoRA weights are normalized, multiplied by the input hidden state, and scaled by a shared magnitude vector before aggregation with the backbone and shared expert outputs, while an expert selection loss sharpens activation sparsity.

Our contributions are threefold:

1. We propose FURINA, the first, to our knowledge, fully mergeable MoE–LoRA architecture. Unlike existing MoE-LoRA methods, FURINA can be seamlessly re-parameterized into a single LoRA adapter or directly into the backbone LLM after training. This ensures full compatibility with high-performance inference frameworks like vLLM and introduces **zero additional latency or complexity** during deployment.

2. We introduce a novel Self-Routing mechanism that eliminates the need for a discrete router via three key innovations: (a) decoupling the learning of direction and magnitude in adapters, (b) a shared magnitude vector for uniform activation scaling, and (c) an expert selection loss that promotes specialization. We also propose a shared expert to mitigate the diminished output norm, preserving model capacity.

3. We conduct extensive experiments on multiple LLMs and benchmarks. The results demonstrate that FURINA significantly enhances the performance of standard LoRA and achieves competitive or superior results compared to state-of-the-art non-mergeable MoE-LoRA methods, while eliminating the corresponding inference-time costs.

## 2 PRELIMINARIES

**Low-Rank Adaption** LoRA introduces trainable low-rank matrices to adapt large-scale pre-trained models efficiently. Typically applied to the multi-layer perceptron (MLP) layers, given a frozen weight matrix $W \in \mathbb{R}^{c_2 \times c_1}$ and an input hidden state $x \in \mathbb{R}^{c_1}$, the output of a LoRA-adapted layer is formulated as:

$$y = Wx + BAx, \tag{1}$$

where $A \in \mathbb{R}^{r \times c_1}$ and $B \in \mathbb{R}^{c_2 \times r}$ are learnable low-rank matrices with rank $r \ll \min(c_1, c_2)$.

**MoE–LoRA** The MoE paradigm has been integrated with LoRA to enhance model capacity while maintaining parameter efficiency. This approach initializes $N$ distinct LoRA adapters $(B_i, A_i)_{i=1}^{N}$ and employs a router function $\phi$ to select a sparse combination of $K$ experts for each input. The router is typically implemented as a trainable gating network. For input $x$, the routing weights are computed as

$$\phi(x)_i = \begin{cases} \frac{1}{Z} r(x)_i, i \in \arg \underset{i}{\operatorname{top}} K\left(r(x)_i\right), \\ 0, \text{otherwise}, \end{cases} \quad r(x) = \operatorname{Softmax}(W_G x), \tag{2}$$

where $W_G \in \mathbb{R}^{N \times c_1}$ is a learnable projection matrix and $Z$ is a normalization constant. The final output becomes

$$y = Wx + \sum_{i \in U}^{N} \phi(x)_i B_i A_i x, \quad U = \arg \topK_j (r(x)_j), \tag{3}$$

while MoE-LoRA enhances model capacity, it introduces a critical limitation: the discrete routing function $\phi(x)$ prevents the merging of adapters into the backbone model. The adapted weight $W' = W + \sum_i \phi(x)_i B_i A_i$ varies with each input $x$, necessitating separate execution of router and experts during inference. This significantly increases implementation complexity and computational overhead compared to standard LoRA. The proposed FURINA framework addresses this fundamental limitation while preserving the capacity benefits of MoE-LoRA, enabling both enhanced expressivity and efficient deployment.

## 3 METHOD

### 3.1 OVERVIEW OF THE PROPOSED FURINA

As detailed in the previous section, the MoE-enhanced LoRA architecture employs a router to dynamically activate specific LoRA adapters. To emulate this behavior, the proposed FURINA incorporates three key components: (1) Decoupled learning of direction and magnitude for LoRA adapters, (2) Shared learnable magnitude vector for uniform activation scaling, and (3) Expert selection loss that encourages divergent activation like MoE. As depicted in Fig. 2, for each LoRA adapter, we simultaneously calculate the activation of the input hidden state and the norm of the adapter. The activations are multiplied by the reciprocal of the adapter norm for normalization. To avoid the uniform activation, the expert selection loss maximizes the activation share of the top-K LoRA experts. We also introduce shared experts to capture the shared knowledge across FURINA experts. Without a router, the shared experts could be combined with the backbone.

### 3.2 MERGEABLE SELF-ROUTING MECHANISM

Mergeable Self-routing is the most important mechanism of FURINA, enabling the merging of LoRA experts. Since $W$ is frozen, we focus mainly on the adapted part of the output hidden state. For simplicity, we denote the adapted part of the MLP layer for MoE–LoRA as follows:

$$\Delta y = y - Wx = \sum_i \phi(x)_i B_i A_i x. \tag{4}$$

The simplest way to eliminate the routers is to assign the same weight to all experts:

$$\Delta y = \sum_i^N B_i A_i x. \tag{5}$$

However, this formulation will degrade the MoE of LoRA to the naive LoRA, where $B' = (B_1, B_2, \ldots, B_N)$ and $A' = (A_1^T, A_2^T, \ldots, A_N^T)^T$.

**Decoupled Learning** To solve the aforementioned issue, we need to review the original MoE of LoRA. Denote the full-size weight matrix of the $i_{th}$ LoRA adapter as $W_i = B_i A_i$, and its column-wise norm could be calculated as follows:

$$d_i = W_i e, \quad e = \underbrace{[1, 1, \ldots, 1]^T}_{\times c_1}. \tag{6}$$

Denote $\hat{W}_i = \text{diag}(d_i + \epsilon)^{-1} W_i$, in which $\epsilon$ is a small positive number to prevent division of $0$, and diag expands a vector to the corresponding diagonal matrix. The formulation of MoE–LoRA in Eq. 4 could be reformulated as follows:

$$\Delta y = \sum_i^N \underbrace{\text{diag}(d_i + \epsilon)}_{\text{magnitude}} \cdot \underbrace{\phi(x)_i \hat{W}_i \hat{x}}_{\text{reweighted similarity}} \cdot \|x\|, \tag{7}$$

in which $\hat{x} = x / \|x\|$. Similarly, after eliminate the router, Eq. 5 could be reformulated as follows:

$$\Delta y = \sum_i^N \underbrace{\text{diag}(d_i + \epsilon)}_{\text{magnitude}} \cdot \underbrace{\hat{W}_i \hat{x}}_{\text{similarity}} \cdot \|x\|. \tag{8}$$

Different from Eq. 7, eliminating the router creates a problematic coupling between the magnitude and similarity terms. The similarity calculation embeds the inverse of the magnitude without any independent operations. Consequently, when the similarity term is multiplied by the magnitude term, they cancel each other out due to the inverse relationship. This cancellation nullifies the intended effect entirely and disables measuring input-to-adapter similarity via the output norm. To address this, we decouple the magnitude and the similarity by replacing the magnitude with a learnable vector $v_i \in \mathcal{R}^{c_2}$, and approximate the MoE of LoRA adapters as follows:

$$\Delta y = \sum_i^N \mathrm{diag}(v_i)(\hat{W}_i \hat{x} \| x \|) = \sum_i^N \mathrm{diag}(v_i)(\hat{W}_i x). \tag{9}$$

**Shared Magnitude Vector**  Although decoupled learning addresses the issue of magnitude-similarity cancellation, the following issue still exists: the output norm of an expert could be large even if the similarity term is small (indicating the expert should not be activated), when the magnitude $v_i$ is large. To address this, we introduce a shared magnitude vector $v$ for all experts. It is worth noting that we do not need to calculate the full-size $\hat{W}$ and $\mathrm{diag}(v)$ during training. The output could be further reformulated as follows:

$$\Delta y = \sum_i^N \mathrm{diag}(v)(\hat{W}_i x) = \sum_{i=1}^N \left( v \otimes \frac{1}{d_i + \epsilon} \right) \otimes B_i A_i x, \tag{10}$$

in which $\otimes$ represents element-wise multiplication. Note that $\tilde{B}_i$ is only calculated once per batch.

**Shared Experts**  Unlike standard MoE–LoRA methods that normalize routing weights to enforce exactly $K$ active experts, cross-expert re-weighting is not applicable in our approach. In extreme cases, the output of the MoE may approach zero, thereby limiting its learning capacity. To mitigate this issue, we introduce a shared expert component to FURINA. Specifically, the incremental output $\Delta y$ with shared expert is calculated as follows:

$$\Delta y = \sum_{i=1}^{N_{SE}} B_i A_i x + \sum_{j=N_{SE}+1}^N \mathrm{diag}(v) \hat{W}_i x. \tag{11}$$

## 3.3 TRAINING OBJECTIVES

The training objective consists of two parts: (1) the supervised fine-tuning (SFT) loss, and (2) the expert selection loss. The SFT loss, similar to the prior approaches, is defined as follows:

$$\mathcal{L}_{\mathrm{SFT}} = \frac{1}{|y|} \sum_t \mathrm{CE}(f(y_t | x, y_{<t})), \tag{12}$$

in which CE represents the cross-entropy loss.

**Expert Selection Loss**  We propose the expert selection loss to encourage the self-routing of LoRA adapters to approximate the function of routers. Specifically, denote the activations of the $i_{th}$ LoRA adapter (apart from the shared experts, if any) as $\mathbf{a}_i$. To encourage divergent activation per token, we introduce the divergence loss as follows:

$$\mathcal{L}_{\mathrm{div}} = -\log \left( \frac{\sum_{i \in \mathcal{S}} |\mathrm{sum}(\mathbf{a}_i)|}{\sum_j |\mathrm{sum}(\mathbf{a}_j)|} \right), \mathcal{S} = \arg \mathrm{top} K_k \left( \left| \sum_j \mathbf{a}_{k,j} \right| \right), \mathbf{a}_i = \hat{W}_i x, \tag{13}$$

in which $\mathcal{S}$ demonstrates the indices of the selected experts. Moreover, we also introduce the balance loss of expert selection. Specifically, given a batch of logits $\mathcal{X} \in \mathbb{R}^{B,T,N}$ in which $B, T, N$ demonstrate the batch size, number of tokens per sample, and the total number of experts. First, we calculate the activation frequency of each expert:

$$\mathcal{F}_i = \left( \sum_u \sum_v \mathbb{I}(i \in \mathcal{S}_{u,v}) \right) \Big/ T, \mathcal{P}_i = \sum_u \sum_v |\mathcal{X}_{u,v,i}| \Big/ T, \tag{14}$$

in which $\mathcal{S}_{u,v}$ represents the selected expert of token $\mathcal{X}_{u,v,:}$, $\mathbb{I}(\cdot) = 1$ if the input condition is "True", otherwise it equals to 0. Then the balance loss could be calculated as follows:

$$\mathcal{L}_{\mathrm{bal}} = N \times \sum_i \mathcal{F}_i \mathcal{P}_i. \tag{15}$$

Denote it as $\mathcal{L}_{\mathrm{bal}}$, given a certain batch of input, the expert selection loss $\mathcal{L}_{sel}$ is defined as their summation. The overall training objective is defined as follows:

$$\mathcal{L} = \mathcal{L}_{\mathrm{SFT}} + \alpha \mathcal{L}_{\mathrm{sel}}, \quad \mathcal{L}_{\mathrm{sel}} = \mathcal{L}_{\mathrm{div}} + \mathcal{L}_{\mathrm{bal}}, \tag{16}$$

in which $\alpha$ represents the loss coefficient, which is set to 0.01 in our work.

### 3.4 MERGING OF EXPERTS

During inference, unlike the standard MoE–LoRA approaches, the multiple LoRA adapters of the proposed FURINA could be merged without loss of information. Without loss of generality, we start from the full FURINA to merge all the experts into one LoRA adapter, and furthermore, into the backbone network. First, we re-write Eq. 11 as follows:

$$
\begin{aligned}
\Delta y &= \sum_{i=1}^{N_{SE}} B_i A_i x + \sum_{j=N_{SE}+1}^{N} \mathrm{diag}(v)\hat{W}_j x \\
&= \sum_{i=1}^{N_{SE}} B_i A_i x + \sum_{j=N_{SE}+1}^{N} \mathrm{diag}(v) \otimes \frac{1}{d_j + \epsilon}(B_j A_j x) \\
&= \sum_{i=1}^{N_{SE}} B_i A_i x + \sum_{j=N_{SE}+1}^{N} \mathrm{diag}\left(v \otimes \frac{1}{d_j + \epsilon}\right) B_j A_j x \\
&= \sum_{i=1}^{N_{SE}} B_i A_i x + \sum_{j=N_{SE}+1}^{N} \tilde{B}_j A_j x.
\end{aligned}
\tag{17}
$$

Then we could merge all these LoRA adapters and the backbone as follows:

$$
\begin{cases}
\mathbf{B} = (B_1, B_2, \ldots, B_{N_{SE}}, \tilde{B}_{N_{SE}+1}, \ldots, \tilde{B}_N), \\
\mathbf{A} = (A_1^T, A_2^T, \ldots, A_N^T)^T.
\end{cases}
\tag{18}
$$

Then the output could be formulated as:

$$
y + \Delta y = Wx + \mathbf{B}\mathbf{A}x = (W + \mathbf{B}\mathbf{A})x.
\tag{19}
$$

For FURINA without shared experts, $N_{SE} = 0$, $\mathbf{B}$ could be reformulated as follows:

$$
\mathbf{B} = (\tilde{B}_1, \tilde{B}_2, \ldots, \tilde{B}_N).
\tag{20}
$$

## 4 EXPERIMENTS

### 4.1 IMPLEMENTATION DETAILS

For each model, we leverage 7 benchmarks to evaluate the capacity of the PEFT methods, including CSQA (Talmor et al. (2019)), HellaSwag (Zellers et al. (2019)), Winogrande (Sakaguchi et al. (2021)), ARC-c and ARC-e (Clark et al. (2018)), OBQA (Mihaylov et al. (2018)), and BoolQ (Clark et al. (2019)). Details of the benchmarks are provided in the Appendix. Three different LLMs are included in our experiment: Qwen2-7B (Yang et al. (2024)), OpenChat-8B (Wang et al. (2024)), and LLaMA3.2-3B, covering different architectures and model scales. Following the setting in MixLoRA (Li et al. (2024)), the learning rate, the rank of each LoRA adapter $r$, the number of experts $N$, and the number of activated experts $K$ are set to $2 \times 10^{-4}$, 16, 8, and 2, respectively. For our proposed FURINA method, we include one shared expert alongside the self-routed experts (maintaining eight experts in total). All experiments are conducted on Nvidia GPUs.

### 4.2 COMPARISON WITH SOTA APPROACHES

We compare FURINA against state-of-the-art PEFT methods, including mergeable baselines (LoRA, DoRA) and non-mergeable MoE-LoRA approaches (MixLoRA, LoRAMoE, MoLA). We measure Time to First Token (TTFT) and latency using Llama3.2-3B. Mergeable methods (LoRA, DoRA, FURINA) are evaluated using vLLM+EvalScope, while non-mergeable methods use MoE-PEFT. For a fair comparison under equivalent parameter budgets, we set LoRA and DoRA rank to 128, matching the total rank of MoE methods. As shown in Tab. 2, FURINA significantly improves upon standard LoRA (+4.8% average gain) and achieves competitive performance with MoE-LoRA methods. Meanwhile, FURINA maintains inference latency equivalent to standard LoRA, with detailed settings and results in Appendix A.3.

Table 2: Comparison with LoRA-styled PEFT approaches on downstream tasks. We employ **blue** and **bold** to indicate the best and second-best results for each model and the average performance. † represents FURINA without shared experts. ‡ represents that the method is not applicable to vLLM, thus evaluated on the MoE-PEFT framework. FURINA achieves competitive performance with standard MoE–LoRA methods with much less computational overhead.

| Method | AVG | OpenChat-8B | Llama3.2-3B | Qwen2-7B | TTFT (ms, ↓) | Latency (ms, ↓) |
|---|---|---|---|---|---|---|
| Single Adapter LoRA | | | | | | |
| LoRA | 78.8 | 80.6 | 77.4 | 78.5 | $\approx 10$ | $\approx 800$ |
| DoRA | 77.1 | 79.0 | 77.9 | 74.3 | | |
| Standard MoE of LoRA | | | | | | |
| MixLoRA | 83.1 | 84.1 | 79.3 | **85.8** | $\approx 550^{\ddagger}$ | $\approx 9000^{\ddagger}$ |
| LoRAMoE | **83.5** | 86.1 | 79.1 | 85.3 | $\approx 450^{\ddagger}$ | $\approx 6000^{\ddagger}$ |
| MoLA | 83.4 | 84.3 | 80.3 | 85.6 | $\approx 700^{\ddagger}$ | $\approx 14500^{\ddagger}$ |
| Fully Mergeable MoE of LoRA | | | | | | |
| **FURINA$^{\dagger}$ (Ours)** | 81.1 | 86.1 | 73.9 | 83.3 | $\approx 10$ | $\approx 800$ |
| **FURINA (Ours)** | 83.6 | 85.3 | 79.7 | 85.9 | | |

### 4.3 COMPATIBILITY WITH LORA VARIATIONS

We also evaluate the compatibility of FURINA with existing LoRA variations, including LoRA+ (Hayou et al. (2024)) and rsLoRA (Kalajdzievski (2023)). For LoRA+, we maintain a learning rate ratio of 5.0 between matrices $B$ and $A$. As shown in Tab. 3, FURINA is consistently compatible with these variants, achieving superior performance compared to standard MoE-LoRA approaches.

Table 3: Compatibility of FURINA with LoRA variations

| Method | OpenChat-8B | Llama3.2-3B | Qwen2-7B | AVG |
|---|---|---|---|---|
| LoRA | 80.6 | 77.4 | 78.5 | 78.8 |
| LoRAMoE | 86.1 | 79.1 | 85.3 | 83.5 |
| SLIM | 87.4 | 79.2 | 85.9 | 84.2 |
| **FURINA** | 85.3 | 79.7 | 85.9 | 83.6 |
| **FURINA w/ LoRA+** | 87.2 | 81.3 | 86.9 | 85.1 |
| **FURINA w/ rsLoRA** | 87.6 | 81.2 | 86.9 | 85.2 |

### 4.4 ABLATION STUDY

**Ablations on the Main Modules** We conduct the ablation study on the main modules proposed in our work. We include Qwen2-7B and LLaMA3.2-3B with all benchmarks, apart from the HellaSwag (because of its size), in the main ablation study. "Decoupled Learning" demonstrates the column normalization of the LoRA adapters, and "Shared Magnitude Vector" represents the introduction of the shared magnitude vector $v$ after normalization. The result shown in Tab. 4 demonstrates that normalizing the LoRA adapters significantly improves the model performance by 3.4%, representing the importance of balanced effectiveness of each expert. The introduction of shared experts and shared magnitude vector also boosts the model performance by 1.4%, demonstrating the benefit of learning mutual foundation knowledge. Further mimicking the activation pattern of standard MoE by $\mathcal{L}_{\text{sel}}$ boosts the model by 0.3%, indicating the importance of emulating the top-K pattern of MoE.

**Effect of Number of Shared Experts** We conduct an experiment on the OBQA and CSQA datasets to validate the influence of the number of shared experts. We average the performance of

Table 4: Ablation study of the main modules. The performance (Perf.) is the average of Qwen2-7B and LLaMA-3.2-3B.

| Decoupled Learning | Shared Experts | Shared Magnitude Vector | $\mathcal{L}_{\text{sel}}$ | Perf (%) |
|:---:|:---:|:---:|:---:|:---:|
| × | × | × | × | 75.9 |
| ✓ | × | × | × | 79.3 |
| ✓ | ✓ | × | × | 80.2 |
| ✓ | ✓ | ✓ | × | 80.7 |
| ✓ | ✓ | ✓ | ✓ | **81.0** |

Figure 3: The effect of the number of shared experts (a) and the scale of the loss coefficient $\alpha$ (b)

LLaMA-3.2-3B and Qwen2-7B. The result is demonstrated in Fig. 3(a). Validation across different LLMs demonstrates that setting the number of shared experts to 1 is sufficient. Further increasing it may result in model performance degradation and convergence to the original LoRA approach.

**Effect of Loss Coefficient** We also conduct an experiment to validate the influence of the loss coefficient $\alpha$. The result is demonstrated in Fig. 3(b). A large $\alpha$ results in over-focus on the imitation of the expert selection patterns of MoE models and limits the learning capacity of FURINA. On the contrary, a minor $\alpha$ cannot guide the LoRA adapters of the MoE to mimic the selective expert activation pattern of standard MoE architectures. Experiments demonstrate that setting $\alpha$ to the range of $[1e-2, 1e-3]$ could achieve an optimal trade-off for these two factors.

**Comparison of Shared Magnitude Vector with Decoupling LoRA Adapters** To validate the utilization of the shared magnitude vector instead of simply decouple the direction and magnitude of the LoRA adapters, we conduct an experiment on the OBQA and CSQA datasets and the LLaMA3.2-3B model. The results are shown in Tab. 5. Utilizing the shared magnitude vector achieves higher performance with fewer trainable parameters. This may be because, although utilizing different magnitude vectors for the adapters increases the number of trainable parameters, it results in an imbalance in the activation of the adapters.

Table 5: Comparison of shared magnitude vector with decoupling of LoRA direction and magnitude

| Method | OBQA | CSQA | AVG |
|:---:|:---:|:---:|:---:|
| Decoupling | 82.4 | 78.5 | 80.4 |
| **Shared Mag. Vec. (Ours)** | **84.0** | 78.5 | **81.2** |

**Evaluation on GSM8K and HumanEval Datasets** To validate the model capacity on reasoning tasks, we also conduct an experiment on the GSM8K (Cobbe et al. (2021)) and HumanEval (Chen et al. (2021)) datasets. We keep the hyperparameters as in the main experiments, and all models

are fine-tuned for 1k steps. For the HumanEval dataset, the model is trained on the CodeAlpaca (Chaudhary (2023)) dataset, and the metric is set to Pass@1. The Evalscope framework is adopted for evaluation. The results in Tab. 6 demonstrate that, compared to LoRA, the proposed FURINA could significantly boost the model performance, indicating its generalizability.

Table 6: Comparison of LoRA and FURINA on GSM8K and HumanEval datasets

| Method | OpenChat-8B | Llama-3.2-3B | Qwen2-7B |
|---|---|---|---|
| LoRA | 61.1 | 47.8 | 62.5 |
| **FURINA (Ours)** | **65.1** | **53.4** | **65.7** |

## 5 RELATED WORKS

**LoRA-style PEFT**   LoRA (Hu et al. (2022)) has emerged as a prominent method for PEFT of large language models. Unlike prompt tuning or adapter-based approaches, LoRA's key advantage lies in its mergeability: after training, the low-rank adapters can be consolidated into the original pre-trained weights, introducing zero additional inference latency. DoRA (Liu et al. (2024b)) decouples weight updates into magnitude and direction components, while rsLoRA (Kalajdzievski (2023)) introduces a scaling mechanism to improve training stability. Hayou et al. (2024) identifies an imbalance in learning dynamics between the $A$ and $B$ matrices and proposes differentiated learning rates to address it. Recently, several works have integrated mixture-of-experts (MoE) architectures with LoRA-style PEFT. MixLoRA (Li et al. (2024)) incorporates the sparse MoE structure from Mixtral into LoRA, and MoLA (Gao et al. (2025)) advocates for layer-wise expert configurations with dynamic expert counts. SLIM (Han et al. (2025)) further enhances this approach by blending identity mappings with LoRA adapters to better preserve pre-trained knowledge. A fundamental limitation of these MoE-LoRA methods, however, is their inability to merge experts into the base model post-training, resulting in persistent inference overhead and increased deployment complexity compared to standard LoRA.

**Mixture of Experts Architecture**   MoE has gained significant traction for scaling large language models efficiently. Mixtral 8×7B (Jiang et al. (2024)) stands as the first widely adopted open-source MoE LLM, demonstrating that sparse expert activation can substantially improve parameter efficiency without compromising performance. DeepSeek MoE (Dai et al. (2024)) introduces shared experts that remain active across all inputs to capture common knowledge, thereby enhancing the model's representational capacity. DeepSeek-V3 (Liu et al. (2024a)) further increases the number of shared experts and incorporates a loss-free balancing mechanism to improve training stability. Recent innovations in MoE architectures also explore MoE frameworks for improved computational efficiency. Jin et al. (2025) proposes incorporating non-computational experts—such as identity or constant-output layers—into the MoE framework, reducing inference costs while maintaining model performance. Lv et al. (2025) proposes to remove the routers from MoE, but still retains the non-linear cross-expert operations that prevent merging of experts.

## 6 CONCLUSION

In this work, we propose FURINA, a novel fully mergeable MoE–LoRA framework that overcomes the deployment complexity and overhead of the standard MoE–LoRA while matching and even surpassing their learning capacity. FURINA introduces a Mergeable Self-Routing mechanism with three key components: (1) Decoupled learning of adapter direction and magnitude, (2) Shared magnitude vector for uniform scaling, and (3) Expert selection loss that promotes sparse expert activation. These elements collectively ensure that each expert's output norm reflects its relevance to the input, enabling mergeable routing. We also incorporate shared experts within the MoE–LoRA block that provides essential foundational knowledge so that the other experts can focus on specific tasks. Comprehensive experiments on 9 datasets and 3 LLMs demonstrate that FURINA significantly enhances standard LoRA performance while achieving competitive results compared to SOTA MoE-LoRA methods, while overcoming their complexity and overhead during deployment, and plug-and-play with mainstream deployment frameworks.

REPRODUCIBILITY STATEMENT

We have included the necessary information to reproduce the results reported in our manuscript. The code base is attached to this manuscript in the supplementary material, and we plan to open-source the code and checkpoints upon publication.

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

# A  APPENDIX

## A.1  DETAILS OF THE INVOLVED DATASETS

Table 7: Summary of datasets used in the experiments.

| Dataset | Task Type | Train | Dev | Test |
|---|---|---|---|---|
| OpenBookQA (OBQA) | Multiple-choice QA | 4,957 | 500 | 500 |
| BoolQ | Binary QA | 9,400 | 3,200 | 3,200 |
| HellaSwag | Sentence completion | 40,000 | — | 10,000 |
| WinoGrande (Debiased) | Commonsense reasoning | 9,248 | 1,267 | 1,767 |
| CommonsenseQA (CSQA) | Commonsense QA | 9,798 | 1,224 | 1,225 |
| ARC-c | Commonsense reasoning | 1,418 | — | 1,172 |
| ARC-e | Commonsense reasoning | 2,821 | — | 2,376 |
| GSM8K | Mathematics | 7,200 | — | 1,300 |
| HumanEval | Code generation | — | — | 164 |
| Code Alpaca | Code generation | 20,000 | — | — |

All the mentioned datasets are open-sourced and allow academic use. We report results for the test set when the ground truth is available. Otherwise, we use the dev set.

## A.2  COMPARISON ON HOLD OUT DATASETS

We compare the catastrophic forgetting resulting from LoRA and FURINA. N/A represents the original model. We fine-tune the Llama3.2-3B model with different PEFT approaches on the OBQA dataset, and evaluate on the hold-out datasets, GSM8K and MMLU. The result demonstrates that, compared with LoRA, FURINA could significantly mitigate catastrophic forgetting. Note that the "Rel. Perf. Drop" is calculated as follows:

$$\text{Rel. Perf. Drop} = \frac{1}{N} \sum_{i=1}^{N} \frac{\text{ACC}_i^0 - \text{ACC}_i}{\text{ACC}_i^0}, \tag{21}$$

in which $\text{ACC}_i^0$ and $\text{ACC}_i$ represent the performance of the original and the fine-tuned model on task $i$, respectively.

## A.3  INFERENCE TIME OF PEFT APPROACHES

To evaluate the inference latency of different PEFT methods, we conduct a controlled comparison using a fixed sequence length of 100 input and 100 output tokens. Time to First Token (TTFT) measures the latency until the first output token is generated, while total latency refers to the time cost for generating the entire output sequence. The evaluation is performed across two distinct frameworks to assess both unmerged and merged deployment scenarios:

Table 8: Comparison on out-of-domain dataset. The model is trained on the OBQA dataset.

| Method | GSM8K(%) | MMLU(%) | Rel. Perf. Drop (%,↓) |
|---|---|---|---|
| N/A | 74.7 | 55.8 | N/A |
| LoRA | 60.4 (14.3↓) | 49.7 (6.1↓) | 10.1 |
| **FURINA** | 74.5 (0.2↓) | 52.6 (3.1↓) | **2.0** |
| **FURINA** w/ rsLoRA | 73.5 (1.2↓) | 50.6 (5.2↓) | 3.7 |
| **FURINA** w/ LoRA+ | 73.5 (1.2↓) | 51.5 (4.3↓) | 3.1 |

1. **MoE-PEFT Framework**: This framework is used to evaluate standard MoE-LoRA approaches directly. For a fair comparison, single-adapter LoRA methods are also evaluated in this framework without merging the adapters into the backbone model. Our proposed FURINA method is merged into the standard LoRA architecture under this framework.

2. **vLLM Framework**: This setup represents a production-ready deployment environment. In this scenario, adapters are merged into the backbone model prior to inference, which is only feasible for mergeable PEFT approaches, such as vanilla LoRA and our FURINA. The router-based MoE-LoRA baselines cannot be evaluated in this setting as they are fundamentally unmergeable due to their routing mechanisms.

All prompts and hyperparameters are aligned across the compared approaches. To account for potential variance in time measurements, each test is repeated 5 times per model, with reported results representing the averaged values. Our results in Tab. 9 demonstrate that FURINA achieves comparable inference latency to standard LoRA while significantly outperforming conventional MoE-LoRA methods. This confirms that FURINA maintains the efficiency benefits of mergeable PEFT methods while delivering the performance advantages of MoE approaches.

Table 9: Inference time of different PEFT approaches on different implementation frameworks. [†] represents FURINA without shared experts.

| Method | MoE-PEFT | | vLLM | |
|---|---|---|---|---|
| | TTFT(ms) | Latency(ms) | TTFT(ms) | Latency(ms) |
| Single Adapter LoRA | | | | |
| LoRA | ≈350 | ≈3500 | ≈10 | ≈800 |
| DoRA | ≈400 | ≈9000 | ≈10 | ≈800 |
| Router-based MoE of LoRA | | | | |
| MixLoRA | ≈550 | ≈9000 | N/A | N/A |
| LoRAMoE | ≈450 | ≈6000 | N/A | N/A |
| MoLA | ≈700 | ≈14500 | N/A | N/A |
| SLIM | ≈650 | ≈17000 | N/A | N/A |
| Router-free MoE of LoRA | | | | |
| **FURINA**[†] **(Ours)** | ≈350 | ≈3500 | ≈10 | ≈800 |
| **FURINA (Ours)** | ≈350 | ≈3500 | ≈10 | ≈800 |

## A.4 COMPARISON WITH DIRECTLY ELIMINATING THE ROUTERS OF STANDARD MOE–LORA

To evaluate the effectiveness of the proposed FURINA, we also compare FURINA with directly eliminating the routers during inference. We utilize SLIM as the baseline and merge the adapters as:

$$\mathbf{B} = \frac{1}{N}(B_1, B_2, \ldots, B_N), \mathbf{A} = (A_1^T, A_2^T, \ldots, A_N^T)^T. \tag{22}$$

We utilize $\frac{1}{N}$ to synthesize the re-weighting operation of the routers. The experiment is conducted on Llama3.2-3B. The results are demonstrated in Tab. 10. Directly eliminating the routers from the MoE–LoRA could significantly decrease its performance, even worse than the original model.

Table 10: Effect of directly eliminating routers from MoE–LoRA approaches during inference

| Method | Original Model | SLIM | SLIM w/ merge | FURINA |
|---|---|---|---|---|
| Perf. (%) | 67.5 | 79.2 | 37.4 (41.8↓) | **79.7** |

### A.5 ACTIVATION PATTERN OF LoRA AND FURINA

In Fig. 4, we also compare the activation pattern of FURINA and LoRA. Note that we compare the normalized activation of both approaches on OBQA dataset, with the Llama-3.2-3B model. The result demonstrates that, compared with LoRA, FURINA achieves significantly larger normalized activations, indicating that the proposed approach could effectively increase the angular similarity of input and LoRA weights.

### A.6 HYPER-PARAMETERS OF EVALUATION ON EVALSCOPE.

To validate the fine-tuned models on GSM8K and HumanEval datasets with the EvalScope framework in the main paper, the hyper-parameters are set as in Tab. 11.

Table 11: Hyper-parameters for evaluating GSM8K and HumanEval

| #max tokens | temperature | # shots |
|---|---|---|
| 2048 | 0.0 | 0 |

### A.7 USE OF LLM

We utilize LLM only to refine the writing of this manuscript.

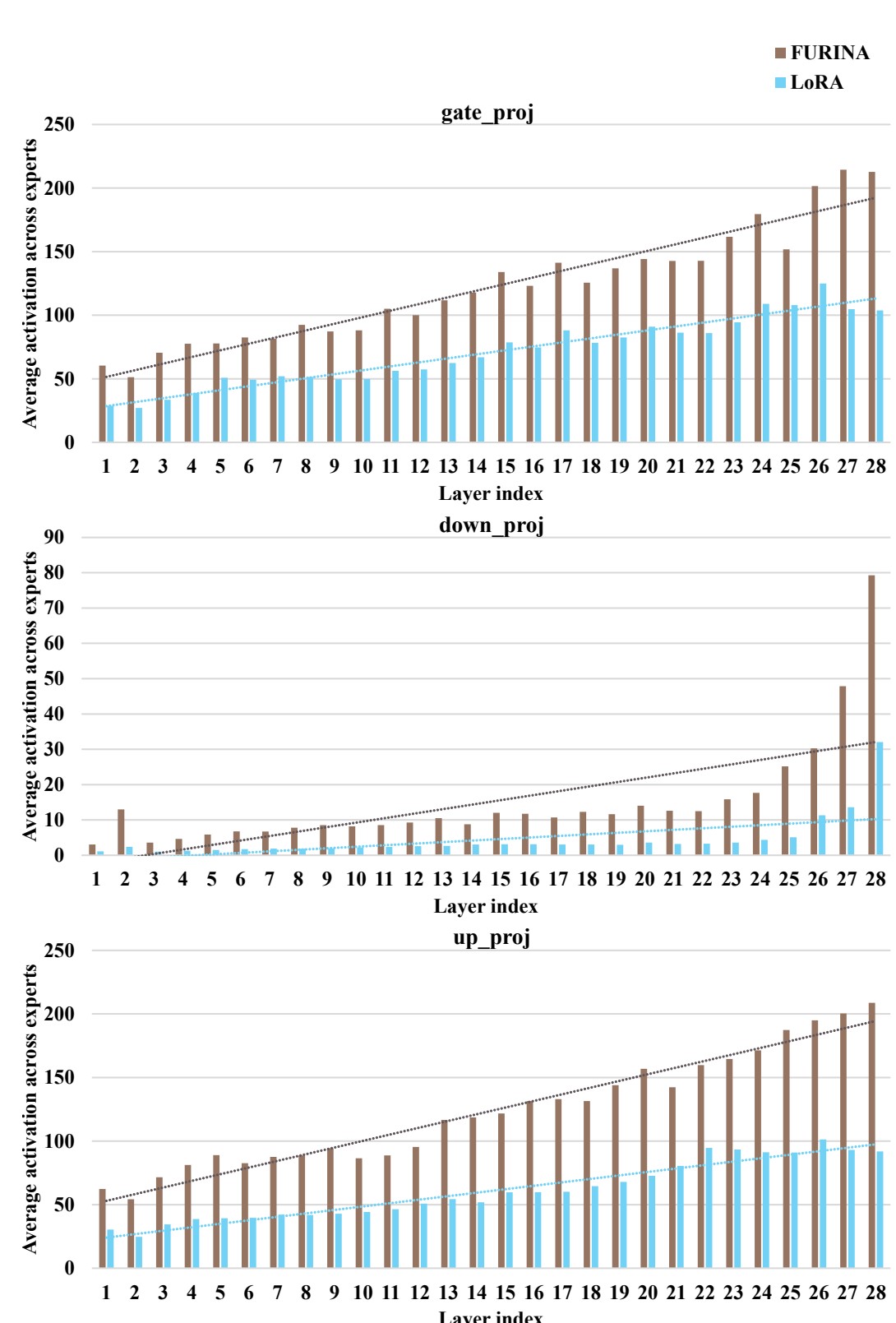

Figure 4: Comparison of activation pattern of FURINA and LoRA across layers

