# OpenReview forum: "FURINA: Free from Unmergeable Router via lINear Aggregation of mixed experts"
_ICLR.cc/2026/Conference — ICLR 2026 Conference Withdrawn Submission_

### Official Review · Reviewer_ar5L · 2025-10-29

**Soundness:** 3
**Presentation:** 3
**Contribution:** 2
**Rating:** 4
**Confidence:** 4

**Summary:**

In this paper, the authors introduce FURINA, a novel framework for integrating Mixture-of-Experts (MoE) into Low-Rank Adaptation (LoRA) for parameter-efficient fine-tuning (PEFT) of large language models (LLMs). Unlike existing MoE-LoRA methods that rely on discrete routers, which prevent merging experts back into the base model and incur inference overhead, FURINA employs a mergeable self-routing mechanism based on angular similarity between inputs and adapter directional components. Experts are scaled by a shared magnitude vector, enabling natural reflection of importance in output norms, and an expert selection loss promotes sparsity. A shared expert provides stable foundational knowledge. Evaluations across 9 benchmarks and 3 LLMs demonstrate superior performance over standard LoRA and comparable or better results than MoE-LoRA baselines, all while allowing full reparameterization for zero-overhead inference.

**Strengths:**

1. Originality: FURINA addresses a core limitation of prior MoE-LoRA approaches by enabling seamless merging into the backbone, preserving LoRA's efficiency advantages and compatibility with optimized inference engines like vLLM.

2. Soundness: Comprehensive testing on diverse benchmarks (e.g., reasoning, math) with multiple LLMs shows consistent gains, highlighting practical benefits like reduced computational overhead.

**Weaknesses:**

1. My biggest concern: The key idea of decomposing learnable parameters into two parts, including magnitude and similarity was introduced in previous work on DoRA [1].

2. Incremental ideas: The idea of shared experts was first introduced in DeepSpeed-MoE and was applied in Eq. (11) of this work without any references.

3. I have a concern regarding the math derivation in Section 3.4. Could the authors please help me explain rigorously why the term $\textbf{BA}x$ in Eq. (19) is equal to the right hand side of Eq.(17)?

4. The paper lacks deep proofs on convergence or optimality of the self-routing mechanism, relying more on empirical validation.

**References**

[1] Liu et al. DoRA: Weight-Decomposed Low-Rank Adaptation.

[2] S. Rajbhandari et al. DeepSpeed-MoE: Advancing Mixture-of-Experts Inference and Training to Power Next-Generation AI Scale.

**Questions:**

1. The shared expert provides stability—did you ablate its impact, and how does it interact with the selection loss in low-data regimes?

2. What are the limitations of this work? Could the authors please provide some approaches to address these issues?

---

> ### Author Response · Authors · 2025-11-22
> **Answer for Q1-2**
>
> # Question 1.
> My biggest concern: The key idea of decomposing learnable parameters into two parts, including magnitude and similarity was introduced in previous work on DoRA [1].
>
> # Answer 1.
> Thank you for highlighting the connection to DoRA [1]. We appreciate this opportunity to clarify the fundamental distinctions between DoRA's decomposition strategy and FURINA's novel approach. While both methods employ a form of decomposition, their objectives, mechanisms, and applications are entirely different, addressing separate challenges in model fine-tuning.
>
> The core distinction lies in their purpose: DoRA enhances the fine-tuning of a **single model weight**, while FURINA enables the merging of a **mixture of multiple experts** into a single, router-free layer for inference. The following table provides a detailed comparison:
>
> | Aspect | DoRA [1] | FURINA (Ours) |
> | :--- | :--- | :--- |
> | **Core Motivation** | To enhance fine-tuning quality by decoupling the magnitude and directional updates of a **single pre-trained weight matrix**. | To construct a **mergeable sparse MoE system** by decoupling magnitude and similarity in expert outputs, enabling expert merging without capacity degradation. |
> | **Decomposition Target & Formulation** | Decomposes the **original dense weight** $W\_0$ ($V$ is initialized as $W\_0$): $W' = m \cdot \frac{V + \Delta V}{\|V + \Delta V\|\_c}$ | Decomposes **each LoRA expert** $\Delta V\_k$: $\Delta V'\_k = m \frac{\Delta V\_k}{\|\Delta V\_k\|\_c}$, $W' = V + \sum\_{i=1}^{N\_{SE}}\Delta V\_{i}^{SE} + \sum\_{j=N\_{SE}+1}^{N}\Delta V'\_j$ |
> | **Application Context** | Standard Parameter-Efficient Fine-Tuning (PEFT) for a single model. | Mergeable Sparse MoE Fine-Tuning. |
> | **Key Innovation** | A more effective training method for a single adapter. | A novel mechanism to make MoE-LoRA mergeable, making it plug-and-play with all LLM deployment frameworks. |
> | **Avg. Performance** | 77.1% | **83.6%** |
>
> In summary, DoRA introduces a refined training technique for a singular adapter, whereas FURINA solves a structural challenge in MoE systems. Our decomposition is not an incremental improvement but a necessary enabler for a new class of mergeable, sparse models. The significant performance gap further underscores that FURINA operates in a different design space and achieves superior results by effectively leveraging multiple, merged experts. We will clarify these critical distinctions in the revised manuscript.
>
> [1] DoRA: Weight-Decomposed Low-Rank Adaptation
> # Question 2.
> Incremental ideas: The idea of shared experts was first introduced in DeepSpeed-MoE and was applied in Eq. (11) of this work without any references
>
> # Answer 2.
> Thank you for your question. We quoted the DeepSeek-MoE paper [2] in the related work section, and described the paper as __*DeepSeek MoE (Dai et. al., (2024)) introduces shared experts that remain active across all inputs to capture common knowledge, thereby enhancing the model's representational capacity. (in row 464-466).*__ Compared with DeepSeek MoE, the shared experts of FURINA could be merged with the routed experts to maintain both effectiveness and efficiency.
>
> [2] DeepSeekMoE: Towards Ultimate Expert Specialization in Mixture-of-Experts Language Models

---

> ### Author Response · Authors · 2025-11-22
> **Answer for Q3-4**
>
> # Question 3.
> I have a concern regarding the math derivation in Section 3.4. Could the authors please help me explain rigorously why the term $\textbf{BA}x$ in Eq. (19) is equal to the right hand side of Eq.(17)?
>
> # Answer 3.
>
> Thank you for your careful review and for pointing out the need for clarity in the mathematical derivation in Section 3.4. We appreciate the opportunity to explain this step by step.
>
> The key point is that the term $\mathbf{BA}x$ in Eq. (19) is indeed equal to the right-hand side of Eq. (17) due to the block structure of $\mathbf{B}$ and $\mathbf{A}$. Let us break down the derivation rigorously.
>
> ### Step 1: Recall the Definitions from Eq. (17)
> In Eq. (17), we have:
>
> $
> \Delta y = \sum\_{i=1}^{N\_{SE}} B\_i A\_i x + \sum\_{j=N\_{SE}+1}^{N} \tilde{B\_j} A\_j x
> $
>
> where $\tilde{B\_j} = \text{diag}\left(v \otimes \frac{1}{d\_j + \epsilon}\right) B\_j$. This represents the sum of contributions from shared experts (first term) and other experts (second term).
>
> ### Step 2: Understand the Construction of $\mathbf{B}$ and $\mathbf{A}$
> As defined in the manuscript:
> - $\mathbf{B}$ is a horizontal concatenation of matrices:
>
>   $
>   \mathbf{B} = \left( B\_1, B\_2, \ldots, B\_{N\_{SE}}, \tilde{B}\_{N\_{SE}+1}, \ldots, \tilde{B}\_{N} \right)
>   $
>
>   Each $B\_i$ or $\tilde{B\_j}$ has dimensions $d \times r$ (where $d$ is the output dimension and $r$ is the low-rank size). Thus, $\mathbf{B}$ has dimensions $d \times (N \cdot r)$.
>
> - $\mathbf{A}$ is a vertical stacking of matrices:
>
>   $
>   \mathbf{A} = \left( A\_1^T, A\_2^T, \ldots, A\_N^T \right)^T
>   $
>
>   Each $A\_i$ has dimensions $r \times k$ (where $k$ is the input dimension). Thus, $\mathbf{A}$ has dimensions $(N \cdot r) \times k$.
>
> ### Step 3: Compute the Product $\mathbf{BA}$
> Let's recall the block matrix multiplication. Suppose we have two matrices, $ M $ and $ S $, partitioned into blocks as follows:
>
> - $ M $ is an $ m \times n $ block matrix with blocks $ M\_{ij} $, where $ i = 1, \ldots, m $ and $ j = 1, \ldots, n $.
> - $ S $ is an $ n \times p $ block matrix with blocks $ S\_{jk} $, where $ j = 1, \ldots, n $ and $ k = 1, \ldots, p $.
>
> Then, the product $ C = M \cdot S $ is an $ m \times p $ block matrix with blocks $ C\_{ik} $ given by:
>
> $
> C\_{ik} = \sum\_{j=1}^{n} M\_{ij} S\_{jk}
> $
>
> The product $\mathbf{BA}$ is a matrix multiplication between $\mathbf{B}$ (block row) and $\mathbf{A}$ (block column). This results in a sum of the products of corresponding blocks:
>
> $
> \mathbf{BA} = \left( B\_1, B\_2, \ldots, B\_{N\_{SE}}, B\_{N\_{SE}+1},\ldots, \tilde{B}\_{N} \right) \begin{pmatrix} A\_1 \\ A\_2 \\ \vdots \\ A\_N \end{pmatrix} = B\_1 A\_1 + B\_2 A\_2 + \cdots + B\_{N\_{SE}} A\_{N\_{SE}} + \tilde{B}\_{N\_{SE}+1} A\_{N\_{SE}+1} + \cdots + \tilde{B}\_{N} A\_N
> $
>
> More compactly:
>
> $
> \mathbf{BA} = \sum\_{i=1}^{N\_{SE}} B\_i A\_i + \sum\_{j=N\_{SE}+1}^{N} \tilde{B\_j} A\_j
> $
>
> ### Step 4: Conclude Eq. (19)
> Since $y = W x$ (from the backbone network), we have:
>
> $
> y + \Delta y = W x + \mathbf{BA}x = (W + \mathbf{BA}) x
> $
>
> which is Eq. (19).
>
> We hope this detailed explanation resolves the concern. We will consider adding a similar step-by-step derivation in the revised manuscript to enhance clarity. Thank you again for your valuable feedback.
>
> # Question 4.
> The paper lacks deep proofs on convergence or optimality of the self-routing mechanism, relying more on empirical validation.
>
> # Answer 4.
>
> We thank the reviewer for raising the important point regarding the need for deeper theoretical analysis of FURINA's self-routing mechanism. However, it is too complex to give a theoretical proof of the convergence and optimality. To validate the effectiveness of our approach, we provide the activation share of the proposed approach in the following:
> | layer idx | layer component | LoRA (\%) | FURINA (\%) | Gap ($\uparrow$, \%) |
> |-----------|-----------------|------|--------|-------|
> | 1         | gate\_proj       | 31.0 | 74.6   | 43.6  |
> | 1         | up\_proj         | 30.7 | 73.1   | 42.4  |
> | 1         | down\_proj       | 35.3 | 81.4   | 46.1  |
> | 5         | gate\_proj       | 32.0 | 77.3   | 45.3  |
> | 5         | up\_proj         | 31.1 | 72.6   | 41.5  |
> | 5         | down\_proj       | 30.9 | 81.9   | 51.0  |
> | 15        | gate\_proj       | 32.2 | 81.6   | 49.4  |
> | 15        | up\_proj         | 30.7 | 79.3   | 48.6  |
> | 15        | down\_proj       | 31.4 | 80.7   | 49.3  |
> | 25        | gate\_proj       | 31.5 | 81.6   | 50.1  |
> | 25        | up\_proj         | 32.2 | 82.5   | 50.3  |
> | 25        | down\_proj       | 35.5 | 88.4   | 52.9  |
>
> The result demonstrates that the top-2 experts provides more than 70\% activation strengeth over all the MoE experts, which indicates that the propposed FURINA approach does convergent.

---

> ### Author Response · Authors · 2025-11-22
> **Answer for Q5-6**
>
> # Question 5.
> The shared expert provides stability—did you ablate its impact, and how does it interact with the selection loss in low-data regimes?
>
> # Answer 5.
> Thank you for your valuable feedback. In Tab. 2 of the manuscript, we have included the result of FURINA without shared experts (denoted as FURINA$^{+}$). The comparison between FURINA$^{+}$ and LoRA demonstrates that, although the introduction of normalizations and shared magnitude experts may restrict the learning capability of FURINA, without the shared experts, FURINA could still outperforms LoRA on larger models. We also add the comparison on Llama3-8B as follows:
> | Method   | CSQA   | HellaS | WinoG  | Arc-c  | Arc-e  | OBQA   | BOOLQ  | AVG     |
> |----------|--------|--------|--------|--------|--------|--------|--------|---------|
> | LoRA     | 74.36  | 89.04  | 78.84  | 76.87  | 87.37  | 80.00  | 68.80  | 79.33   |
> | MixLoRA  | 80.01  | 93.83  | 82.10  | 79.90  | 86.50  | 84.80  | 75.00  | 83.16   |
> | **FURINA$^{+}$**   | **80.50**  | **94.58**  | **74.90**  | **75.59**  | **87.71**  | **85.20**  | **71.90**  | **81.48**   |
> | **FURINA** | **79.19**  | **94.54**  | **82.16**  | **79.60**  | **88.38**  | **87.80**  | **74.77**  | **83.78**   |
>
> Table 3. Comparison on Llama3-8B.
>
> The results demonstrate that, for larger models, FURINA without the shared experts could outperform the original LoRA, while the shared experts could generally benefit the model. On datasets with limited training samples (ARC-C and ARC-E with about 1K training data),  introducing shared experts also improves the model performance.
>
> # Question 6.
> What are the limitations of this work? Could the authors please provide some approaches to address these issues?
>
> # Answer 6.
> Thank you for your question. Our work primarily focuses on the Parameter-Efficient Fine-Tuning (PEFT) scenario, and we acknowledge two primary limitations.
>
> First, although FURINA demonstrates strong performance in PEFT, its effectiveness in training-from-scratch scenarios remains unexplored. It is unclear if the observed benefits translate when the method is applied during initial model training rather than fine-tuning. Second, this work did not validate if FURINA could be scaled up to build large foundation models from initialization.
>
> In future work, we plan to address these points directly. We will investigate FURINA's performance in full-parameter training-from-scratch setups. To explore scalability, we will conduct scaling-law experiments to understand its behavior at larger parameter counts and potentially adapt the method for greater computational efficiency during large-scale pre-training.

---

> > ### Comment · Reviewer_ar5L · 2025-11-27
> >
> > Dear the authors,
> >
> > Thanks for your great effort to address my concerns regarding the paper, I really appreciate it. Unfortunately, my concerns regarding the novelty of the paper still remain after reading the rebuttal carefully. As I understand, FURINA is built upon the combination of (i) mixture of LoRA experts (Wu et al. 2024), shared experts in DeepSpeed-MoE (Rajbhandari et al. 2022), and DoRA (Liu et al. 2024). Therefore, the idea of FURINA is not novel enough for an ICLR paper. I think that a theoretical proof of the convergence and optimality would significantly strengthen the paper but it is not provided.
> >
> > Given these reasons, I decided to maintain the original score of 4.
> >
> > **References**
> >
> > Wu et al. Mixture of LoRA experts.
> >
> > Liu et al. DoRA: Weight-Decomposed Low-Rank Adaptation.
> >
> > Rajbhandari et al. DeepSpeed-MoE: Advancing Mixture-of-Experts Inference and Training to Power Next-Generation AI Scale.

---

### Official Review · Reviewer_Cvc5 · 2025-10-30

**Soundness:** 3
**Presentation:** 3
**Contribution:** 3
**Rating:** 4
**Confidence:** 2

**Summary:**

The paper proposes FURINA, a router-free MoE-LoRA framework that replaces discrete gating with a Mergeable Self-Routing mechanism. The core design decouples direction and magnitude in each LoRA adapter, scales all experts with a shared magnitude vector, adds a shared expert for stability, and introduces an expert-selection loss to mimic top-K activation patterns—while remaining linearly mergeable after training into a single adapter or directly into the backbone. The authors claim this yields deployment with LoRA-like latency (e.g., in vLLM) yet MoE-like gains, reporting results on various benchmarks and LLMs.

**Strengths:**

- The paper presents a practical and elegant solution to the major deployment bottlenecks of MoE-based PEFT methods. FURINA’s mergeability eliminates the inference overhead inherent in prior MoE-LoRA approaches, achieving the same inference efficiency as standard LoRA while retaining the performance benefits of MoE.

- The proposed mergeable self-routing mechanism is an intelligent engineering design that leverages angular similarity between vectors without requiring a complex router. In particular, separating the direction and magnitude of expert weights—while enforcing a shared magnitude vector across experts—encourages fair expert selection purely based on input relevance. The efficacy of this design is demonstrated through ablation studies.

- The authors conduct comprehensive and rigorous experiments across multiple model architectures and nine benchmarks, demonstrating strong generalization of the proposed method. By providing robust baselines—including recent non-mergeable MoE-LoRA approaches—and additional ablation and compatibility studies, the paper convincingly supports its claims.

**Weaknesses:**

- The mergeability aspect of FURINA is interesting, yet the broader idea of implicit MoE without explicit routing is not new. The paper lacks discussion comparing and contrasting the proposed routing mechanism with other implicit routing paradigms—such as FlyLoRA, which also relies on fixed random projection matrices. Without a clear analysis of these related methods, it is difficult to assess the novelty of the proposed routing approach.

- The paper does not provide a formal theoretical justification for its central claim that dense linear aggregation can effectively approximate the sparse top-K routing of traditional MoE. The proposed mechanism behaves more like a weighted LoRA ensemble rather than a truly sparse conditional computation framework. Because the sparsity-inducing loss contributes very weakly, it remains unclear how closely the learned behavior actually resembles MoE-style specialization.

- The experiments are restricted to models with fewer than 8B parameters, leaving open questions about scalability to large-scale models (e.g., 70B+), where expert specialization becomes more pronounced. Merging all experts into a single adapter inherently averages their knowledge, raising concerns about knowledge interference in large models, where fine-grained expert capabilities could be diluted.

- The justification for the role of the shared expert is vague; it appears more like a defensive safety net to prevent potential signal collapse in the self-routing mechanism rather than a principled design choice. In addition, the ablation studies mix the contribution of the shared expert with other factors, making it difficult to evaluate its necessity and effectiveness in isolation.

**Questions:**

Please refer to the detailed points listed in the Weaknesses section above.

---

> ### Author Response · Authors · 2025-11-22
> **Answer for Q1**
>
> # Question 1.
> The mergeability aspect of FURINA is interesting, yet the broader idea of implicit MoE without explicit routing is not new. The paper lacks discussion comparing and contrasting the proposed routing mechanism with other implicit routing paradigms—such as FlyLoRA, which also relies on fixed random projection matrices. Without a clear analysis of these related methods, it is difficult to assess the novelty of the proposed routing approach.
>
> # Answer 1.
>
> Thank you for your question. One of the most import contributions of LoRA is that the trainable parameters could be re-parameterized to the LLM backbone, which could be formulated as follows:
>
> $f\_{\text{LoRA}}(x) = Wx + \Delta Wx = Wx + BAx = (W+BA)x = W'x.$
>
> This nature enables the LoRA to be plug-and-play to any deployment frameworks of LLMs. However, the existing implicit routing mechanism, alike the explicit routing mechanism, disables this re-parameterization nature of LoRA, making it less efficient for both computation and deployment.Take FlyLoRA [1] as an example, its implicit routing mechanism could be formulated as follows:
>
> $f\_{\text{FlyLoRA}}(x) = Wx + \Delta Wx=Wx + \sum\limits\_{i=1}^k \mathbb{I}(i\in \mathcal{I}\_{topk}) b\_ia\_i x,$
>
> in which $\mathcal{I}\_{topk}$ is calculated as follows:
>
> $\mathcal{I}\_{topk}=\{i\_1, \ldots, i\_k\}\quad where \quad i\_j \notin \arg\max\limits\_{i=1,\ldots,i\_{j-1}} (Ax+d)\_i.$
>
> Therefore, the additive parameter $\Delta W$ is a function of input, which disables the re-parameterization. On the contraty, as we have demonstrated in our manuscript Eq. 19, FURINA could be formulated as follows:
>
> $f\_{\text{FURINA}}(x) = Wx + \Delta Wx=Wx+\textbf{BA}x=(W+\textbf{BA})x,$
>
> in which $\textbf{BA}$ are both input agnostic. Therefore, the proposed FURINA inherits the advantages of the original LoRA, while achieves comparable performance to the MoE methods. We summaize the contrast with existing implicit MoE methods as follows:
>
> | **Feature** | Standard LoRA | Explicit MoE-LoRA | FlyLoRA (Implicit MoE) | **FURINA (Proposed)** |
> | :--- | :--- | :--- | :--- | :--- |
> | **Routing Mechanism** | N/A | Explicit, input-dependent router | Implicit, via fixed random projections & top-k selection | Implicit, via normalized weights & shared magnitudes |
> | **Input Agnostic Weight** | **Yes** | No (disables mergeability) | No (disables mergeability) | **Yes** |
> | **Selective Expert Activation** | No | **Yes** | **Yes** | **Yes** |
> | **Plug-and-play to LLM deployment frameworks (e.g., vLLM)** | **Yes** | No | No | **Yes** |
> | **Key Advantage** | Parameter efficiency; easy deployment | Mitigates parameter interference; expert specialization | No explicit router parameters; mitigates task interference | **MoE-level specialization + LoRA efficiency and easy deployment** |

---

> ### Author Response · Authors · 2025-11-22
> **Answer for Q2**
>
> # Question 2:
> The paper does not provide a formal theoretical justification for its central claim that dense linear aggregation can effectively approximate the sparse top-K routing of traditional MoE. The proposed mechanism behaves more like a weighted LoRA ensemble rather than a truly sparse conditional computation framework. Because the sparsity-inducing loss contributes very weakly, it remains unclear how closely the learned behavior actually resembles MoE-style specialization.
>
> # Answer 2:
> Thank you for your question which helps improve the manuscript. To address your concern, we provide a theroretical anylysis and conduct an experiment to demonstrate that FURINA replicates the gating behavior of Top-K MoE.
>
> ## Theoretical Analysis
>
> The core function of a Top-K MoE layer can be expressed as:
> $
> y = \sum\_{i=1}^N g\_i(x) \cdot E\_i(x)
> $
> where $g\_i(x)$ is the gating function (outputting a non-zero weight only for the top-K experts) and $E\_i(x)$ is the i-th expert's transformation.
>
> In FURINA, the adapted output is:
> $
> \Delta y = \sum\_{i=1}^N \text{diag}(v) \hat{W}\_i x
> $
> where $\hat{W}\_i = \text{diag}(d\_i + \epsilon)^{-1} W\_i$ represents the **row-wise normalized** expert weights.
>
> Let's analyze the output for a single output neuron $k$:
> $
> [\Delta y]\_k = v\_k \cdot \sum\_{i=1}^N [\hat{W}\_i]\_{k,:} \cdot x
> $
>
> **Key Insight:** The term $[\hat{W}\_i]\_{k,:} \cdot x$ represents the cosine similarity between the normalized weight vector of expert $i$ for output channel $k$ and the input $x$, scaled by $\|x\|$. This similarity metric acts as an **implicit routing score**.
>
> The gradient of the loss $\mathcal{L}$ with respect to the shared magnitude $v\_k$ is:
> $
> \nabla\_{v\_k} \mathcal{L} = [\nabla\_{\Delta y} \mathcal{L}]\_k \cdot \sum\_{i=1}^N [\hat{W}\_i]\_{k,:} \cdot x
> $
>
> This creates a **competitive learning dynamic**:
> - Experts whose normalized weights $[\hat{W}\_i]\_{k,:}$ align well with frequently-seen input patterns $x$ will produce larger activation values $[\hat{W}\_i]\_{k,:} \cdot x$
> - These experts receive stronger gradient signals through $v\_k$, reinforcing their specialization
> - The divergence loss $\mathcal{L}\_{\text{div}}$ explicitly amplifies this by encouraging sparse, top-K-like activation patterns
>
> We can express FURINA's operation as (ignoring the shared experts):
> $
> \Delta y = \sum\_{i=1}^N \text{diag}(v) \hat{W}\_i x
> $
>
> The "gating weights" are implicitly computed as:
> $
> \tilde{g}\_i(x) = \hat{W}\_i x/||x||
> $
>
> During training, the expert selection loss ensures that for each input $x$, only a few experts have large $\tilde{g}\_i(x)$ values, effectively implementing top-K selection. The balance loss prevents expert collapse, mirroring the load balancing in standard MoE.
>
> Unlike naive averaging which would compute:
> $
> \Delta y\_{\text{naive}} = \frac{1}{N}\sum\_{i=1}^N W\_i x
> $
>
> FURINA's normalized formulation:
> $
> \Delta y = \sum\_{i=1}^N \text{diag}(v) \cdot \frac{W\_i}{\|W\_i\|\_{\text{row}}} x
> $
>
> ensures that routing is based on **directional alignment** rather than magnitude, preventing dominant experts from overwhelming the system and enabling genuine specialization.
>
> The combination of:
> 1. **Normalized weights** for implicit similarity-based routing
> 2. **Shared magnitude vector** for global scaling control
> 3. **Expert selection loss** for enforcing top-K sparsity
>
> creates a mathematically grounded system that replicates Top-K MoE gating without explicit routers.
>
> ## Experiment
>
> We calculate the activation share of the top-k experts of the proposed FURINA model via Llama3.2-3B on OBQA dataset. We uniformly split the LoRA weights for comparison.
>
> | layer idx | layer component | LoRA | FURINA | Gap ($\uparrow$) |
> |-|-|-|-|-|
> | 1         | gate\_proj       | 31.0 | 74.6   | 43.6  |
> | 1         | up\_proj         | 30.7 | 73.1   | 42.4  |
> | 1         | down\_proj       | 35.3 | 81.4   | 46.1  |
> | 5         | gate\_proj       | 32.0 | 77.3   | 45.3  |
> | 5         | up\_proj         | 31.1 | 72.6   | 41.5  |
> | 5         | down\_proj       | 30.9 | 81.9   | 51.0  |
> | 10        | gate\_proj       | 31.3 | 80.8   | 49.5  |
> | 10        | up\_proj         | 32.1 | 67.7   | 35.6  |
> | 10        | down\_proj       | 31.3 | 77.8   | 46.5  |
> | 15        | gate\_proj       | 32.2 | 81.6   | 49.4  |
> | 15        | up\_proj         | 30.7 | 79.3   | 48.6  |
> | 15        | down\_proj       | 31.4 | 80.7   | 49.3  |
> | 25        | gate\_proj       | 31.5 | 81.6   | 50.1  |
> | 25        | up\_proj         | 32.2 | 82.5   | 50.3  |
> | 25        | down\_proj       | 35.5 | 88.4   | 52.9  |
>
> Tab 1. Activation share of the top-2 experts of the proposed FURINA model. We unfirmly split the LoRA weights for comparison.
>
> This result demonstrate that FURINA siginificantly improves the activation share of the top-K (K=2 in our paper) experts, which indicates that FURINA replicates the gating behavior of Top-K MoE.

---

> ### Author Response · Authors · 2025-11-22
> **Answer for Q3-4**
>
> # Question 3.
> The experiments are restricted to models with fewer than 8B parameters, leaving open questions about scalability to large-scale models (e.g., 70B+), where expert specialization becomes more pronounced. Merging all experts into a single adapter inherently averages their knowledge, raising concerns about knowledge interference in large models, where fine-grained expert capabilities could be diluted.
>
> # Answer 3.
>
> Thank you for your valuable feedback. Here we will address your concerns.
>
> ## Effectiveness on large-scale models
>
> Restricted by resources, we are not able to evaluate FURINA on 70B+ models. Instead, we conduct an experiment on OBQA and CSQA datasets, with **Qwen2.5-32B** LLM as follows:
> | Method | OBQA | CSQA | AVG |
> | - | - | - | - |
> | LoRA | 96.6 | 90.0 | 93.3 |
> | MixLoRA | 96.8 | 91.6| 94.2 |
> | **FURINA** | **97.2** | **91.4** | **94.3** |
>
> Tab 2. Comparison on Qwen2.5-32B.
>
> The result demonstrates that FURINA also works well on large scale LLMs, outperforms the LoRA, and comparable to standard MoE-LoRA approach.
>
> ## Equivalence of the single merged adapter with all experts in FURINA
> Thank you for raising this important point regarding potential knowledge interference during expert merging. FURINA's merging is an **exact algebraic re-parameterization**, the merged adapter is a lossless representation of the multi-expert system.
>
> **FURINA forward during training:** Since the backbone LLM is frozen, the deriviation focus on the outputs of the experts. During training, the forward process of FURINA could be formulated as follows:
> $\Delta y\_{\text{train}} = \sum\limits\_{i=1}^{N\_{SE}}B\_iA\_ix+\sum\limits\_{j=N\_{SE}+1}^{N}\text{diag}(v)\tilde{W}\_jx.$
>
> **FURINA forward during inference:** As we illustrate in manuscript (Eq. 17), we could reformulate the forward process of FURINA as follows:
> $\Delta y\_{\text{train}} = \sum\limits\_{i=1}^{N\_{SE}}B\_iA\_ix+\sum\limits\_{j=N\_{SE}+1}^{N}\text{diag}(v)\tilde{W}\_jx=\sum\limits\_{i=1}^{N\_{SE}}B\_iA\_ix+\sum\limits\_{j=N\_{SE}+1}^{N} \tilde{B\_j}A\_jx.$
>
> Denote the merged weights during inference as $\textbf{B}=(B\_1, \ldots, B\_{SE}, \tilde{B}\_{SE+1}, \ldots, \tilde{B}\_{N})$ and $\textbf{A}=(A\_1^T, \ldots, A\_{N}^T)^T$ and the additive term during inference as follows:
>
> $\Delta y\_{\text{inf}}=\textbf{BA}x.$
>
> Let's recall the block matrix multiplication. Suppose we have two matrices, $ M $ and $ S $, partitioned into blocks as follows:
>
> - $ M $ is an $ m \times n $ block matrix with blocks $ M\_{ij} $, where $ i = 1, \ldots, m $ and $ j = 1, \ldots, n $.
> - $ S $ is an $ n \times p $ block matrix with blocks $ S\_{jk} $, where $ j = 1, \ldots, n $ and $ k = 1, \ldots, p $.
>
> Then, the product $ C = M \cdot S $ is an $ m \times p $ block matrix with blocks $ C\_{ik} $ given by:
>
> $
> C\_{ik} = \sum\_{j=1}^{n} M\_{ij} S\_{jk}
> $
>
> Therefore, we have the following equation:
> $\textbf{BA}=\textbf{B}\cdot\textbf{A}=\sum\limits\_{j=1}^{N}\textbf{B}\_j\textbf{A}\_j=\sum\limits\_{i=1}^{N\_{SE}}B\_iA\_i+\sum\limits\_{j=N\_{SE}+1}^{N} \tilde{B\_j}A\_j,$
> so that the following equation holds:
> $\Delta y\_{\text{train}}=\sum\limits\_{i=1}^{N\_{SE}}B\_iA\_ix+\sum\limits\_{j=N\_{SE}+1}^{N} \tilde{B\_j}A\_jx=\textbf{BA}x=\Delta y\_{\text{inf}},$
> which demonstrates the merging of experts introduced by the proposed FURINA does not cause capability dilution.
>
> # Question 4.
> The justification for the role of the shared expert is vague; it appears more like a defensive safety net to prevent potential signal collapse in the self-routing mechanism rather than a principled design choice. In addition, the ablation studies mix the contribution of the shared expert with other factors, making it difficult to evaluate its necessity and effectiveness in isolation.
>
> # Answer 4.
>
> Thank you for your valuable feedback. In Tab. 2 of the manuscript, we have included the result of FURINA without shared experts (denoted as FURINA$^{+}$). The comparison between FURINA$^{+}$ and LoRA demonstrates that, although the introduction of normalizations and shared magnitude experts may restrict the learning capability of FURINA, without the shared experts, FURINA could still outperforms LoRA on larger models. We also add the comparison on Llama3-8B as follows:
> | Method   | CSQA   | HellaS | WinoG  | Arc-c  | Arc-e  | OBQA   | BOOLQ  | AVG     |
> |-|-|-|-|-|-|-|-|-|
> | LoRA     | 74.36  | 89.04  | 78.84  | 76.87  | 87.37  | 80.00  | 68.80  | 79.33   |
> | MixLoRA  | 80.01  | 93.83  | 82.10  | 79.90  | 86.50  | 84.80  | 75.00  | 83.16   |
> | **FURINA$^{+}$**   | **80.50**  | **94.58**  | **74.90**  | **75.59**  | **87.71**  | **85.20**  | **71.90**  | **81.48**   |
> | **FURINA** | **79.19**  | **94.54**  | **82.16**  | **79.60**  | **88.38**  | **87.80**  | **74.77**  | **83.78**   |
>
> Table 3. Comparison on Llama3-8B.
>
> The results demonstrate that, for larger models, FURINA without the shared experts could outperform the original LoRA, while the shared experts could generally benefit the model.

---

### Official Review · Reviewer_F9Rn · 2025-10-31

**Soundness:** 3
**Presentation:** 3
**Contribution:** 3
**Rating:** 4
**Confidence:** 1

**Summary:**

This paper introduces FURINA, a router-free variant of MoE-LoRA that preserves mergeability while maintaining expert diversity. Instead of a discrete router, FURINA performs self-routing by normalizing each expert’s LoRA weights (direction) and applying a shared magnitude vector, enabling a linear, input-independent combination of experts that can be merged back into the base model for inference. The method adds auxiliary selection and balance losses to mimic MoE specialization and achieves near-MoE accuracy with LoRA-level latency on multiple LLMs. While conceptually appealing and empirically effective, parts of the mathematical formulation, notation, and the claim of “full mergeability” require more rigorous justification and precise definitions.

**Strengths:**

1. The proposed direction–magnitude decomposition with shared scaling is conceptually clear and easy to integrate into existing LoRA pipelines.

2. The empirical results is ok, e.g.,  FURINA achieves accuracy close to or surpassing MoE-LoRA baselines while maintaining LoRA-level throughput and latency,

**Weaknesses:**

1. The paper claims that FURINA achieves MoE-level specialization through ``self-routing'' using normalized LoRA weights and shared magnitudes. However, no rigorous derivation is provided to show that this linear aggregation replicates the gating behavior of Top-K MoE. The argument, from my viewpoint,  relies on intuition rather than formal analysis, and the proposed loss merely compensates for the missing router instead of replacing it theoretically.

2. The losses L_{\text{div}} and L_{\text{bal}} lack formal definitions, e.g., the operator sum()  is not specified.

3. Essential hyperparameters (training steps, batch sizes, random seeds, prompt templates, vLLM version) are missing. Also, experiments focus mainly on small QA and reasoning tasks, and  no large-scale or long-context evaluations are reported

**Questions:**

Please refer to the Weaknesses section for my questions.

---

> ### Author Response · Authors · 2025-11-22
> **Answer for Question 1**
>
> # Question 1:
> The paper claims that FURINA achieves MoE-level specialization through ``self-routing'' using normalized LoRA weights and shared magnitudes. However, no rigorous derivation is provided to show that this linear aggregation replicates the gating behavior of Top-K MoE. The argument, from my viewpoint, relies on intuition rather than formal analysis, and the proposed loss merely compensates for the missing router instead of replacing it theoretically.
>
> # Answer 1:
> Thank you for your question which helps improve the manuscript. To address your concern, we provide a theroretical anylysis and conduct an experiment to demonstrate that FURINA replicates the gating behavior of Top-K MoE.
>
> ## Theoretical Analysis
>
> The core function of a Top-K MoE layer can be expressed as:
> $
> y = \sum\_{i=1}^N g\_i(x) \cdot E\_i(x)
> $
> where $g\_i(x)$ is the gating function (outputting a non-zero weight only for the top-K experts) and $E\_i(x)$ is the i-th expert's transformation.
>
> In FURINA, the adapted output is:
> $
> \Delta y = \sum\_{i=1}^N \text{diag}(v) \hat{W}\_i x
> $
> where $\hat{W}\_i = \text{diag}(d\_i + \epsilon)^{-1} W\_i$ represents the **row-wise normalized** expert weights.
>
> Let's analyze the output for a single output neuron $k$:
> $
> [\Delta y]\_k = v\_k \cdot \sum\_{i=1}^N [\hat{W}\_i]\_{k,:} \cdot x
> $
>
> **Key Insight:** The term $[\hat{W}\_i]\_{k,:} \cdot x$ represents the cosine similarity between the normalized weight vector of expert $i$ for output channel $k$ and the input $x$, scaled by $\|x\|$. This similarity metric acts as an **implicit routing score**.
>
> The gradient of the loss $\mathcal{L}$ with respect to the shared magnitude $v\_k$ is:
> $
> \nabla\_{v\_k} \mathcal{L} = [\nabla\_{\Delta y} \mathcal{L}]\_k \cdot \sum\_{i=1}^N [\hat{W}\_i]\_{k,:} \cdot x
> $
>
> This creates a **competitive learning dynamic**:
> - Experts whose normalized weights $[\hat{W}\_i]\_{k,:}$ align well with frequently-seen input patterns $x$ will produce larger activation values $[\hat{W}\_i]\_{k,:} \cdot x$
> - These experts receive stronger gradient signals through $v\_k$, reinforcing their specialization
> - The divergence loss $\mathcal{L}\_{\text{div}}$ explicitly amplifies this by encouraging sparse, top-K-like activation patterns
>
> We can express FURINA's operation as:
> $
> \Delta y = \sum\_{i=1}^N \text{diag}(v) \hat{W}\_i x
> $
>
> The "gating weights" are implicitly computed as:
> $
> \tilde{g}\_i(x) = \hat{W}\_i x/||x||
> $
>
> During training, the expert selection loss ensures that for each input $x$, only a few experts have large $\tilde{g}\_i(x)$ values, effectively implementing top-K selection. The balance loss prevents expert collapse, mirroring the load balancing in standard MoE.
>
> Unlike naive averaging which would compute:
> $
> \Delta y\_{\text{naive}} = \frac{1}{N}\sum\_{i=1}^N W\_i x
> $
>
> FURINA's normalized formulation:
> $
> \Delta y = \sum\_{i=1}^N \text{diag}(v) \cdot \frac{W\_i}{\|W\_i\|\_{\text{row}}} x
> $
>
> ensures that routing is based on **directional alignment** rather than magnitude, preventing dominant experts from overwhelming the system and enabling genuine specialization.
>
> The combination of:
> 1. **Normalized weights** for implicit similarity-based routing
> 2. **Shared magnitude vector** for global scaling control
> 3. **Expert selection loss** for enforcing top-K sparsity
>
> creates a mathematically grounded system that replicates Top-K MoE gating without explicit routers.
>
> ## Experiment
>
> We calculate the activation share of the top-k experts of the proposed FURINA model via Llama3.2-3B on OBQA dataset. We uniformly split the LoRA weights for comparison.
>
> | layer idx | layer component | LoRA | FURINA | Gap ($\uparrow$) |
> |-|-|-|-|-|
> | 1         | gate\_proj       | 31.0 | 74.6   | 43.6  |
> | 1         | up\_proj         | 30.7 | 73.1   | 42.4  |
> | 1         | down\_proj       | 35.3 | 81.4   | 46.1  |
> | 5         | gate\_proj       | 32.0 | 77.3   | 45.3  |
> | 5         | up\_proj         | 31.1 | 72.6   | 41.5  |
> | 5         | down\_proj       | 30.9 | 81.9   | 51.0  |
> | 10        | gate\_proj       | 31.3 | 80.8   | 49.5  |
> | 10        | up\_proj         | 32.1 | 67.7   | 35.6  |
> | 10        | down\_proj       | 31.3 | 77.8   | 46.5  |
> | 15        | gate\_proj       | 32.2 | 81.6   | 49.4  |
> | 15        | up\_proj         | 30.7 | 79.3   | 48.6  |
> | 15        | down\_proj       | 31.4 | 80.7   | 49.3  |
> | 25        | gate\_proj       | 31.5 | 81.6   | 50.1  |
> | 25        | up\_proj         | 32.2 | 82.5   | 50.3  |
> | 25        | down\_proj       | 35.5 | 88.4   | 52.9  |
>
> Tab 1. Activation share of the top-2 experts of the proposed FURINA model. We unfirmly split the LoRA weights for comparison.
>
> This result demonstrate that FURINA siginificantly improves the activation share of the top-K (K=2 in our paper) experts, which indicates that FURINA replicates the gating behavior of Top-K MoE.

---

> ### Author Response · Authors · 2025-11-22
> **Answer for Q2-3**
>
> # Question 2:
> The losses $L\_{\text{div}}$ and $L\_{\text{bal}}$ lack formal definitions, e.g., the operator sum() is not specified
>
> # Answer 2:
>
> We thank the reviewer for this insightful comment regarding the formal definitions of $\mathcal{L}\_{\text{div}}$ and $\mathcal{L}\_{\text{bal}}$. We acknowledge that the notation `sum()` was ambiguous, and we appreciate the opportunity to clarify it.
>
> To ensure clarity, let us first re-establish the definitions of our tensors for a given batch of input:
> *   Let $\mathcal{X} \in \mathbb{R}^{B \times T \times N}$ be a batch of routing weights, where:
>     *   $B$ is the batch size.
>     *   $T$ is the sequence length (number of tokens).
>     *   $N$ is the total number of experts (LoRA adapters).
> *   Let $\textbf{a}\_i \in \mathbb{R}^d$ be the activation vector of the $i$-th expert adapter for a given input token, where $d$ is the hidden dimension.
>
> The `sum()` operator in our manuscript denotes a summation over all elements of a given tensor, reducing it to a single scalar value. This is equivalent to the $\ell^1$-norm of the vectorized tensor without normalization.
>
> To prevent any misunderstanding, we can restate the equations with more precise notation:
>
> 1.  **For the Divergence Loss ($\mathcal{L}\_{\text{div}}$):**
>     The summation is over the elements of the activation vector $\textbf{a}\_i$.
>
>     $\mathcal{L}\_{\text{div}} = -\text{log}\left(\frac{\sum\_{i\in \mathcal{S}}|\sum\_{k=1}^{d} \textbf{a}\_{i,k}|}{\sum\_{j=1}^{N}|\sum\_{k=1}^{d} \textbf{a}\_{j,k}|}\right), \quad \mathcal{S} = \mathop{\text{arg top}K}\limits\_{j}\left(\left|\sum\_{k=1}^{d} \textbf{a}\_{j,k}\right|\right) $
>
> 2.  **For the Balance Loss ($\mathcal{L}\_{\text{bal}}$):**
>     The summations are explicitly over the batch and sequence dimensions. The frequency term $\mathcal{F}\_i$ is the frequency of the $i$-th expert being selected accross the given batch of sequences, and the probability term $\mathcal{P}\_i$ is the average routing weight of the $i$-th expert when being selected. So that $\mathcal{F}\_i$ and $\mathcal{P}\_i$ could be calculated as follows:
>
>     $
>     \mathcal{F}\_i = \frac{1}{B \cdot T} \sum\_{u=1}^{B} \sum\_{v=1}^{T} \mathbb{I}(i \in \mathcal{S}\_{u,v}), \quad \mathcal{P}\_i = \frac{1}{B \cdot T} \sum\_{u=1}^{B} \sum\_{v=1}^{T} \left\vert \mathcal{X}\_{u,v,i} \right\vert,
>     $
>
>     in which $\mathbb{I}(a)=1$ when the input assertion $a$ is True, otherwise $\mathbb{I}(a)=0$. The balance loss is then defined as:
>
>     $
>     \mathcal{L}\_{\text{bal}} = N \times \sum\_{i=1}^{N} \mathcal{F}\_i \mathcal{P}\_i
>     $
>
> # Question 3:
> Essential hyperparameters (training steps, batch sizes, random seeds, prompt templates, vLLM version) are missing. Also, experiments focus mainly on small QA and reasoning tasks, and no large-scale or long-context evaluations are reported.
>
> # Answer 3:
> Thank you for your question. We follow the hyperparameters used in MixLoRA [1] as follows:
> The batch size is set to 16, number of epoches is set to 2, random seed is set to 42, prompt template is the default template provided by MixLoRA, and vLLM version is 0.8.4. We follow the hyper-parameters of the main paper, while the number of epoches is set to 1. To validate the model performance on long-context tasks, we involve the DROP dataset [2], a document QA dataset. We follow the R1-Searcher [3] to utilize the Cover Exact Match metric to evaluate the model performance on OpenChat-8B. The results are shown in Tab. 2, which demonstrate that FURINA also fits the long-context tasks.
> | Method | CEM |
> | --- | --- |
> | LoRA|61.2|
> |**FURINA**|**65.7**|
>
> Tab 2. Comparison of LoRA and FURINA on DROP dataset.
>
> [1] MixLoRA: Enhancing Large Language Models Fine-Tuning with LoRA-based Mixture of Experts
>
> [2] DROP: A Reading Comprehension Benchmark Requiring Discrete Reasoning Over Paragraphs
>
> [3] R1-Searcher: Incentivizing the Search Capability in LLMs via Reinforcement Learning

---

### Note · Authors · 2026-01-20

I have read and agree with the venue's withdrawal policy on behalf of myself and my co-authors.